# Genetic Diversity and Structure of Rear Edge Populations of *Sorbus aucuparia* (Rosaceae) in the Hyrcanian Forest

**DOI:** 10.3390/plants10071471

**Published:** 2021-07-19

**Authors:** Hamed Yousefzadeh, Shahla Raeisi, Omid Esmailzadeh, Gholamali Jalali, Malek Nasiri, Łukasz Walas, Gregor Kozlowski

**Affiliations:** 1Department of Environmental Science, Faculty of Natural Resources, Tarbiat Modares University (TMU), Mazandaran 14115-111, Iran; 2Department of Forest Science and Engineering, Faculty of Natural Resources, Tarbiat Modares University (TMU), Mazandaran 14115-111, Iran; reisi.shahla@yahoo.com (S.R.); oesmailzadeh@modares.ac.ir (O.E.); jalali_g@modares.ac.ir (G.J.); 3Department of Forestry, Faculty of Natural Resources, Tehran University (TU), Tehran 31587-77871, Iran; nasiri.malek@gmail.com; 4Department of Biogeography and Systematics, Institute of Dendrology, Polish Academy of Sciences, Parkowa 5, PL-62-035 Kornik, Poland; lukaswalas@man.poznan.pl; 5Department of Biology and Botanic Garden, University of Fribourg, Chemin du Musée 10, CH-1700 Fribourg, Switzerland; gregor.kozlowski@unifr.ch; 6Natural History Museum Fribourg, Chemin du Musée 6, CH-1700 Fribourg, Switzerland; 7Eastern China Conservation Centre for Wild Endangered Plant Resources, Shanghai Chenshan Botanical Garden, 3888 Chenhua Road, Songjiang, Shanghai 201602, China

**Keywords:** conservation genetics, inbreeding depression, range-edge populations, rowan tree, Hyrcanian forest

## Abstract

*Sorbus aucuparia* (Rosaceae) is a small tree species widely distributed in Eurasia. The Hyrcanian forest is the southernmost distribution limit of this species. Severe habitat degradation and inadequate human interventions have endangered the long-term survival of this species in this region, and it is necessary to develop and apply appropriate management methods to prevent the loss of its genetic diversity. In this study, we used 10 SSR markers in order to evaluate the genetic diversity of this taxon. Leaf samples were collected from five known populations of *S. aucuparia* throughout its distribution area in the Hyrcanian forest. Expected heterozygosity ranged from 0.61 (ASH) to 0.73, and according to the M-ratio, all populations showed a significant reduction in effective population size, indicating a genetic bottleneck. Global F_ST_ was not statistically significant and attained the same values with and without excluding null alleles (ENA) correction (F_ST_ = 0.12). Bayesian analysis performed with STRUCTURE defined two genetic clusters among the five known populations, while the results of discriminant analysis of principal components (DAPC) identified three distinct groups. The average proportion of migrants was 22. In general, the gene flow was asymmetrical, with the biggest differences between immigration and emigration in Barzekoh and Asbehriseh. The Mantel test showed that there was no significant correlation between genetic distance (F_ST_) and geographic distance in *S. aucuparia*. The best pathway for theoretical gene flow is located across the coast of the Caspian Sea and significant spatial autocorrelation was observed in only one population. In order to reduce the extinction risk of very small and scattered populations of *S. aucuparia* in the Hyrcanian forest, it is very important to establish and/or enhance the connectivity through habitat restoration or genetic exchange.

## 1. Introduction

The Hyrcanian forest, located along the southern coast of the Caspian Sea in Iran and Azerbaijan, is one of the most important biodiversity centers on our planet [1]. The area possesses a remarkable amount of nearly 150 woody species, among them numerous relict trees [2,3]. The main reason of this impressive tree and shrub diversity lies in the fact that this region was never covered by glaciers during the Pleistocene [4,5].

The rowan tree (*Sorbus aucuparia* L.) is one of the most important species of the genus *Sorbus*, which has a medicinal value, and has a wide natural range in areas with low and high altitudes from the Atlantic coasts of Europe to the Kamchatka Peninsula and East China in Asia [6,7,8]. The Hyrcanian forest is the southernmost distribution limit of *S. aucuparia*, with only a few and small populations of this species remaining in this region. The rowan tree is distributed in the Hyrcanian forest at higher altitudes in mountainous regions, reaching the upper forest limit (1800–2800 m a.s.l.) and often growing on rocky slopes [9]. Iranian occurrences of this species are typical rear-edge populations, isolated from each other, and occurring in a scattered distribution. Hence, this species in the Hyrcanian region is vulnerable and highly sensitive to climate change.

Future climate change may alter the genetic diversity within species [10] through the reduction of species distribution and increase of habitat fragmentation [11]. It has been reported by many researchers that rising temperatures and drought stress over the last half-century, increased mortality and decreased the growth of plants. This effect is especially strong in the case of edge populations [12].

Marginal populations are potentially important for conservation, since they may preserve rare alleles and gene combinations important for adaptation to extreme environmental conditions [13,14]. However, the assessment of genetic diversity and evolution of peripheral populations is still insufficient [15]. Hoffmann et al. [16] showed that decreasing adaptation potential to severe conditions is often encountered at range edges [17]. This is connected with increased genetic drift, which leads to a reduction in gene diversity. On the other hand, Sáenz-Romero et al. [18] mentioned that in some cases, the migratory fluxes from core populations may improve genetic diversity in peripheral populations [18]. 

Severe habitat degradation and inadequate human interventions have endangered the survival of many plant species in the Hyrcanian forest [19] and this is even more worrying for marginal species with very low density and abundance. Additionally, the severe habitat conditions (rocky sites with shallow soil and harsh habitat conditions) of *S. aucuparia* in the Hyrcanian forest are responsible for weak regeneration, as there is a strong relationship between habitat quality and genetic diversity [20]. Thus, the long-term survival of this species in the Hyrcanian forest is uncertain. Moreover, due to the rapid degradation of the Hyrcanian forest, it is necessary to apply appropriate management methods to prevent the decline of plant populations, and consequently, the genetic richness of many species, especially those present in the upper forest border due to their higher vulnerability [14].

Knowledge about the levels and patterns of genetic diversity within and between populations is crucial to adopt a good conservation strategy for potentially threatened species [21]. Several molecular techniques have been used as efficient methods for considering the genetic diversity of the genus *Sorbus* [22,23,24,25,26,27]. Simple sequence repeat marker (SSR) is a cross-selective marker and a powerful tool in evaluating diversity levels, phylogenetic relationships, and genetic structure of the genus *Sorbus* [23,28]. This type of marker has been frequently used in the last years due to its co-dominant character and abundance in the plant’s genome [29], and due to the high transferability to the closely related species [29,30].

This study was designated to investigate the genetic diversity and population genetic structure of *S. aucuparia* in its southernmost distribution area, using a set of 10 SSR markers and using plant material covering the whole known natural distribution of this species in Iran. More specifically, we aimed to answer the following specific questions: (1) What is the genetic diversity of *S. aucuparia* within and between its natural populations in Iran? (2) What is the spatial genetic structure of natural populations of *S. aucuparia* in the study area? (3) What is the migration rate and gene flow between the populations of this species? Finally, based on our results, we are discussing the conservation implications and measures needed for the long-term conservation of this species in the Hyrcanian forest.

## 2. Materials and Methods

### 2.1. Sampling, DNA Extraction, and SSR Amplification

Leaf samples were collected from five known populations of *S. aucuparia* throughout its distribution area in the Hyrcanian forest (Table 1). 

Hoebee et al. [23] concluded that trees are very unlikely to be clones if the minimum distance between trees is 30 m. Hence, depending on the population size, 10–28 mature trees were chosen from each population with at least a 50–100 m distance between trees to avoid recurring genotypes [23]. In total, 78 trees were sampled.

DNA was extracted from the leaf tissue using the CTAB methods [31,32] with some modifications [33]. The quantity and quality of the extracted DNA were determined by loading the samples on agarose 1% gel and using spectrophotometry, respectively. In total, a set of 10 polymorphic SSR markers from 15 initially screened SSR markers were selected to detect the genetic variation among populations (Appendix A). Markers were amplified using a DNA Engine Thermal Cycler (Bio-rad, Hercules, CA, USA). The reaction mixtures of 10 μL contained 1× buffer, 0.2 mM dNTPs, 2.5 mM MgCl_2_, 0.2 μM each SSR forward and reverse primer, 30 ng of genomic DNA, and 1 U of Taq polymerase (Thermo Scientific). The PCR program involved an initial denaturation step of 5 min at 94 °C, followed by 30 cycles at 94 °C for 30 s, the appropriate annealing temperature for 30 s, 72 °C for 40 s, and an extension cycle of 1 min at 72 °C. PCR product was run on 8% polyacrylamide gel and dyed with silver nitrate protocol [34]. The multimode bands were coded in the Gel-Pro analyzer 32 software.

### 2.2. Genetic Diversity

The null allele frequencies of each locus were assessed using Microchecker 2.2.3 software [35]. The average number of alleles (A), number of private alleles [36], and the effective number of alleles (Ae) were calculated using the GENEALEX 6.501 software [37], INEst v. 2.0 [38] was used to estimate the expected heterozygosity (He), observed heterozygosity (Ho), and the inbreeding coefficient (FIS), as well as for a bottleneck test [39]. FSTAT was used to estimate allelic richness (Ar). Global and pairwise F_ST_ were estimated using FREENA and tested with bootstrapping over loci [36]. The significance of a deviation from the Hardy–Weinberg equilibrium, including a Bonferroni correction and the estimated frequency of null alleles, were estimated using CERVUS software. 

### 2.3. Genetic Structure

Analysis of molecular variance (AMOVA) among and within populations was performed using GenAlex [37,40]. From AMOVA, the fixation index (F_ST_) and Nm (haploid number of migrants) within the population were obtained. The Bayesian algorithm implemented in STRUCTURE [41] was used to clustering individuals, whereas discriminant analysis of principal components (DAPC; [42] provided an independent, non-Bayesian method. STRUCTURE procedure included 10^5^ MCMC iterations, 10^4^ burn-in, and 10 independent runs with the maximum number of clusters set to K = 6. Evanno’s delta K method from CLUMPAK software was used to choose the best K. Function ‘find.cluster,’ implemented in the adegenet package in R, was used to estimate the optimal number of clusters for the DAPC. Next, the ‘dapc’ function was used to perform this analysis. To estimate the contemporary dispersal patterns and determining the degree of connectivity in populations under study, assignment analysis was done by GENEALEX 6.501 software. In order to infer historical gene flow (Nm) patterns, MIGRATE-N v3.6 [43] was used to estimate the effective population sizes (θ) and mutation-scaled immigration (M) among the stands [44,45]. Four independent runs with different initial seeds were performed and the Bezier approximation for the marginal likelihood was used to test which run has the best fit for the data. Each run consisted of 50,000 sampled parameter values and 5000 recorded steps after a burn-in of 1000 steps. A static heating scheme was used (chains set at 1, 1.5, 3, 10^5^). The software CIRCUITSAPE [46,47] was used for testing how topography could shape the gene flow between populations. The altitude raster was a resistance surface with analyzed populations as nodes.

### 2.4. Mantel Test

Patterns of isolation by distance (IBD; [48]) were investigated using function ‘Mantel test’ with 9999 iterations implemented in R. The matrix of the genetic distance (pairwise F_ST_ with ENA correction) was tested against the matrix of spatial distance between populations created using the program QGIS.

### 2.5. Spatial Autocorrelation

Spatial autocorrelation analysis [49] was performed in GenAlEx [37]. The spatial autocorrelation coefficient (r) was computed using the multilocus genetic distance and the Euclidean distance between individuals.

## 3. Results

### 3.1. Genetic Diversity

Analysis of 10 microsatellite loci in 78 individuals (genets) showed 41 different alleles. The number of different alleles per locus ranged from 3 (MSS5) to 5.83 (MSS9). The values of He and Ho per locus varied from 0.47 (MSS5) to 0.73 (MSS9, SA08) and from 0.23 (MSS9) to 0.82 (MSS1), respectively. The highest and lowest frequency values of null alleles were in MSS9 (0.28) and MSS16 (0.003), respectively. The mean null allele frequency for all examined populations was 0.10 (Appendix A).

Genetic diversity estimates obtained for each population at the genet level are summarized in Table 2. The expected heterozygosity ranged from 0.61 (ASH) to 0.73 (KH), while Ho ranged from 0.45 (NAV) to 0.56 (BAN and KH). The highest Ar value was in KH (4.56) and the lowest in ASH (3.48). Private alleles were observed in the eastern and western populations (BAN and KH).

According to the *M*-ratio, all populations showed a significant reduction in the effective population size, indicating a genetic bottleneck (Table 2). The spatial pattern of He and Ar is demonstrated in Figure 1; the highest values were observed in the eastern population. F_IS_ ranged from 0.16 to 0.31; according to the DIC in all populations under study, inbreeding was not the likely factor of the deviation in the Hardy–Weinberg equilibrium (Table 2).

Global F_ST_ was statistically insignificant and attained the same values with and without ENA correction (F_ST_ = 0.12). This result suggests that the presence of null alleles does not influence the level of differentiation. The pairwise F_ST_ ranged from 0.005 (between BAN and LOM) to 0.08 (between BAN and ASH), indicating a varied level of differentiation among populations (Table 3).

### 3.2. Spatial Genetic Structure and Gene Flow

Bayesian analysis of a genetic structure, performed in STRUCTURE, defined two genetic clusters among the five analyzed populations (Figure 2a). Three populations of BAN, KH, and NAV were grouped as a single cluster, whereas two populations of ASH and LOM were assigned to the second cluster. We used DAPC analysis to increase the validation and support the output of Bayesian clustering. The results of DAPC for K = 3—best K for STRUCTURE—were relatively dissimilar to those obtained with STRUCTURE. The results of DAPC for K = 3 revealed that the BAN population from the eastern part and KH from the western part of the Hyrcanian forest comprised a separated group. However, three populations (NAV, ASH, and LOM) were not assigned to either of the two detected clusters and presented a relatively high admixture (Figure 2b). This result was also confirmed by the population assignment test (Appendix A).

The results of recent migration rates are shown in Figure 3 and Appendix A. The average proportion of migrants was 22.02, suggesting that more individuals than 20 per population may be migrants. However, differences between populations are very strong—in LOM, the number of migrants was 34.9, whereas in BAN and ASH, this value was lower than 20 (15.9 and 14.0, respectively). In general, the gene flow was asymmetrical, with the biggest differences between immigration and emigration in LOM (strong immigration) and ASH (strong emigration). The intensity of gene flow between populations from the western and eastern parts of the Hyrcanian forests was rather low.

The Mantel test showed that there was no significant correlation between genetic distance (F_ST_) and geographic distance in *S. aucuparia* (r = −0.282, *p* = 0.7). However, a resistance analysis made in CIRCUITSCAPE, with elevation as a matrix of resistance against the F_ST_ matrix, indicated that elevation could be a significant barrier for gene flow across the eastern and western range of the species. CIRCUITSCAPE models the connectivity between stands as a landscape resistance distance (isolation-by-resistance). In our analysis, altitude was used as a resistance raster, and paths without topographical barriers were estimated as best ways for a gene flow. The map generated by CIRCUITSCAPE showed the path of high conductance that represents possible pathways of gene flow among populations; theoretical conductance is presented in Figure 4.

The best pathway for theoretical gene flow is located across the coast of the Caspian Sea. Theoretical southern path across the mountains is less probable, because of topographic complexity. Significant spatial autocorrelation was observed only in population KH (Figure 5). Lack of spatial autocorrelation confirms the Mantel test result and suggests that IBD is irrelevant in the studied populations.

## 4. Discussion

Our study revealed that the Hyrcanian populations of *S. aucuparia* in the southern Caspian Sea have higher genetic diversity compared with reported results for other species of the genus *Sorbus* [23,51,52] and even compared with the populations of. *S. aucuparia* in refugial regions of Europe [52]. This is not surprising because the Hyrcanian region was a potential refugium during the last glacial maximum for a wide range of woody taxa [2,53]. An interesting result is the increase in Ho, Ar, and Ap (except the KH population) from the western to the eastern limit distribution of the *S. aucuparia* in Iran. The western population of *S. aucuparia* in the Hyrcanian forest (KH) with a small area and high tree density that is separated about 100 km from the main region of the Hyrcanian forest, showed the highest heterozygosity, allelic richness, and number of private alleles. An appropriate interpretation is that this population is the nearest to the European populations, and it may has acted as a receiver of genes from western and south-eastern Europe, especially from countries with access to the Black Sea (Turkey and Georgia). On the other hand, the BAN population, as the easternmost population of *S. aucuparia* in the Hyrcanian forest, showed high heterozygosity and private alleles. There are several examples where genetic diversity within populations showed an increase towards species distribution margins [54,55]. Kucerova et al. [56] found higher differentiation over central-European populations than those located in southern locations for *S. torminalis*. Additionally, Jankowska-Wroblewska et al. demonstrated that peripheral populations of *S. torminalis* have relatively high levels of genetic diversity [26].

On a global scale, the *S. aucuparia* populations in the Hyrcanian forest are considered as a range-edge population of this species in the Northern Hemisphere. These range-edge populations are isolated from the populations in Europe and are vulnerable to genetic drift [57]. Inbreeding depression, genetic drift, and differentiation of peripheral populations are all exacerbated by persistent reductions in gene flow among small isolated and less dense populations [57,58]. These interpretations contrast strongly with the high levels of individual heterozygosity, suggesting a heavy selection against selfed offspring [59]. All five stands studied had higher expected heterozygosity than observed heterozygosity, resulting in positive inbreeding coefficients. This is contrary to the gametophytic self-incompatibility system of woody Rosaceae [23]. Given that the size and/or density of a population can influence the outcrossing rate of self-compatible plants [60], it seems that harsh habitat conditions, small size, and low tree density, as well as a severe human intervention, has caused, contrary to expectations for the genus *Sorbus* [26], positive inbreeding in *S. aucuparia* populations in the Hyrcanian forest. Additionally, because of their often high levels of heterozygosity, outcrossing trees such as *S. aucuparia* can be disproportionately vulnerable to a reduction in pollen-mediated gene flow, which can mask deleterious recessive alleles that, if expressed, can lead to a reduction in population’s fitness [61].

A bottleneck was detected in the Hyrcanian populations of *S. aucuparia* using the *M*-ratio with positive inbreeding. Genetic drift is inversely related to the effective population size (1/2Ne; [62]) and typically occurs in small populations, where rare and private alleles face a greater chance of being lost. The current populations of *S. aucuparia* in the north of Iran may be the remnants of a large population from the past, which over time, due to low competitiveness with other species, has decreased their density and nested in harsh sites with steep slopes and rocky outcrops. Additionally, habitat disturbance, such as forest fires or logging, could lead to fragmented habitat and influence genetic patterns and structures, local extinctions, and subsequent colonization.

In the periphery of a species range, abiotic and biotic environments may differ from those in the center, and there are likely less suitable habitats [59]. Habitat suitability, the historical colonization, migration pattern, and geographical distance among populations shaped the genetic structure of a species [63]. In this study, the habitat conditions of KH population were completely different from the other sites. *Sorbus aucuparia* is usually found above the timberline and in the rocky and steep habitat of the Hyrcanian forest, while the KH habitat is a dune forest with relatively suitable soil and a much smaller habitat slope than other habitats. This could be the reason for the higher genetic diversity and completely different genetic structure of *S. aucuparia* in this habitat. Due to the distance of at least 500 km of the population of BAN from the other four Hyrcanian populations and possibly the complete cessation of gene flow over time, it has been differentiated from other populations.

## 5. Conclusions

The results of our study demonstrate a positive inbreeding in *S. aucuparia* populations in the Hyrcanian forest, showing evidence of a past bottleneck. To reduce the extinction risk of very small and isolated populations of *S. aucuparia* in this region, there is a need to establish and/or enhance the connectivity between isolated populations through habitat restoration or genetic exchange. In fact, gene movement via seedling could provide a pathway for dispersal and, as a result, greater genetic diversity retention through increased effective population size, reducing the effects of drift [62,64]. To achieve the above-mentioned goals, suitable new areas for afforestation with *S. aucuparia* should be identified to reduce the geographical distance for gene flow among the main populations. Additionally, improving the habitat quality and increasing the density of trees by planting additional seedlings should be used as another alternative to increase connectivity among neighboring trees and reduce the inbreeding depression within the populations.

## Figures and Tables

**Figure 1 plants-10-01471-f001:**
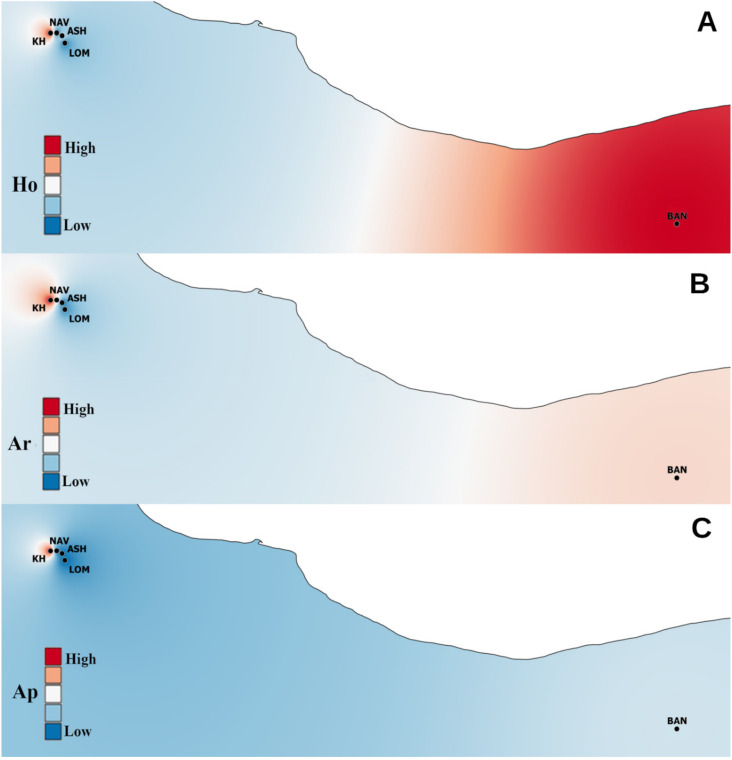
Maps of the genetic diversity of *Sorbus aucuparia* populations visualized by QGIS software. (**A**): expected heterozygosity (Ho), (**B**): allelic richness (Ar), (**C**): number of private alleles (Ap).

**Figure 2 plants-10-01471-f002:**
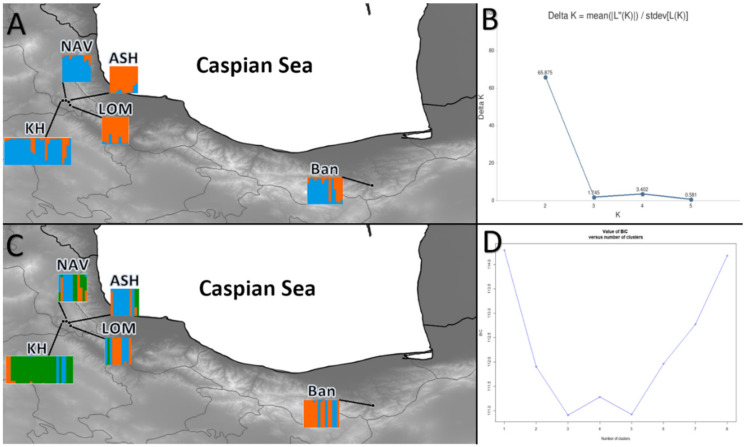
(**A**): Results from STRUCTURE for K = 2 for populations of *Sorbus aucuparia*, (**B**): the optimal number of clusters (K) for STRUCTURE estimated by method from Evanno et al. (2005) [50], (**C**): results from DAPC for K = 3, (**D**): best number of cluster determined by find cluster in R. For abbreviations of the populations, see Table 1.

**Figure 3 plants-10-01471-f003:**
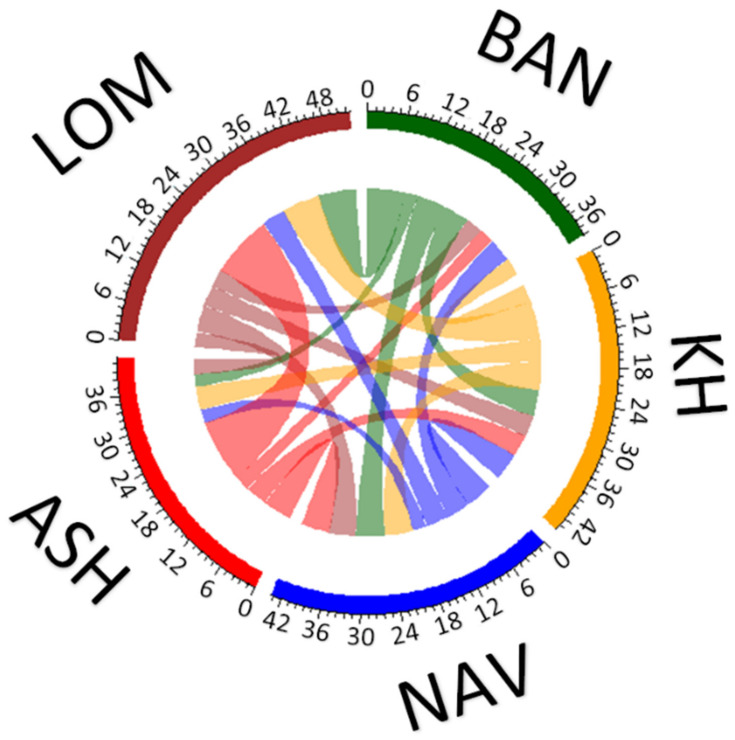
Theoretical gene flow between populations of *Sorbus aucuparia* estimated with MIGRATE-N. For abbreviations of the populations, see Table 1.

**Figure 4 plants-10-01471-f004:**
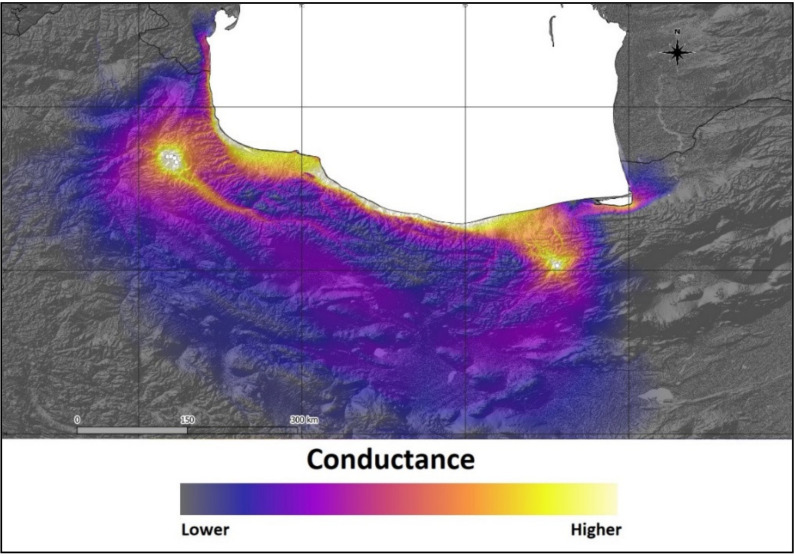
Theoretical gene flow between populations of *Sorbus aucuparia* in relation to the topography determined using CIRCUITSCAPE.

**Figure 5 plants-10-01471-f005:**
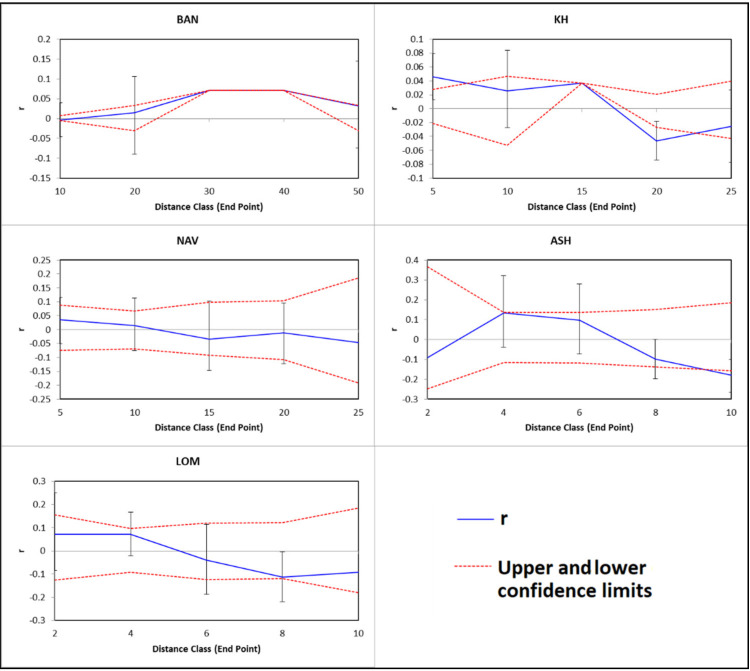
Correlograms illustrating spatial autocorrelation for all analyzed populations. Upper and lower error bars bound the 95% confidence interval to r, as determined by bootstrap resampling.

**Table 1 plants-10-01471-t001:** Geographical characteristics of the studied populations.

Population Name	Sample Code	Longitude	Latitude	Altitude	Sample Size	Associate Species
Khalkhal	KH	373929.7	483540.9	2100–2500	28	*Quercus macranthera-Sorbus graeca*
Olsehposht	NAV	373936.3	484015.2	1650–2100	12	*Fagus orientalis, Acer* spp.
Asbehriseh	ASH	373741.4	484416.2	1500–1900	12	*Carpinus betulus, Acer* spp.
Barzekoh	LOM	373231.5	484634.4	1450–1850	11	*Fagus orientalis, Carpinus orientalis, Acer mazandaranicum*
Sangedeh	BAN	360601.6	531232.5	1900–2400	15	*Betula pendula-Acer hyrcanum*

**Table 2 plants-10-01471-t002:** Parameters of the genetic diversity of the studied populations. For abbreviations of the populations, see Table 1.

Pop	Lat	Long	n	A	Ae	Ar	Ap	Null	Ho	He	Fis	M-Ratio
BAN	36.06	53.124	15	4.56	3.29	4.12	2.00	0.102	0.56	0.70	0.16	0.006
KH	37.393	48.354	28	5.78	3.64	4.56	5.00	0.125	0.56	0.73	0.21	0.050
NAV	37.394	48.401	12	4.56	3.12	4.08	0.00	0.134	0.45	0.69	0.31	0.037
ASH	37.374	48.442	12	3.67	2.79	3.48	0.00	0.082	0.49	0.61	0.17	0.004
LOM	37.323	48.463	11	3.89	2.89	3.61	0.00	0.123	0.46	0.63	0.29	0.000

n—total number of individuals, A—the average number of alleles, Ae—effective number of alleles, Ar—allelic richness, Ap—number of private alleles, Null—frequency of null alleles, Ho—observed heterozygosity, He—expected heterozygosity, F_Is_—fixation index, M-ratio—*p*-value of Wilcox sign-rank test after 10,000 permutations.

**Table 3 plants-10-01471-t003:** Matrix of the genetic distance between populations. For abbreviations of the populations, see Table 1.

	BAN	KH	NAV	ASH	LOM
BAN		0.0337	0.0326	0.0831	0.0057
KH	0.0330		0.0162	0.0655	0.0411
NAV	0.0332	0.0126		0.0779	0.0532
ASH	0.0777	0.0597	0.0732		0.0078
LOM	0.0064	0.0366	0.0527	0.0108	

F_ST_ with ENA correction above the diagonal, F_ST_ without ENA correction below the diagonal.

## Data Availability

Data is attached in Appendix A.

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
