# Peer review of "Genetic Diversity and Structure of Rear Edge Populations of *Sorbus aucuparia* (Rosaceae) in the Hyrcanian Forest"

_plants, 2021, doi:10.3390/plants10071471_

Round 1

Reviewer 1 Report

Paper "Genetic diversity and structure of rear edge populations of Sorbus aucuparia (Rosaceae) in Hyrcanian forest" is very interesting.

All mathematical and statistical methods used in this paper are good.

All tables are perfect.

Figure 3: quality is poor and figure needs correction.

After this correction I recommend this paper to publication in Plants.

Author Response

Answers to reviewers (itemized answers in blue)

Reviewer 1

Paper "Genetic diversity and structure of rear edge populations of Sorbus aucuparia (Rosaceae) in Hyrcanian forest" is very interesting.

Answer: Thank you for this positive comment concerning our study and the manuscript.

All mathematical and statistical methods used in this paper are good.

Answer: Thank you for this positive comment.

All tables are perfect.

Answer: Thank you for this positive comment.

Figure 3: quality is poor and figure needs correction.

Answer: Figure 3 is now replaced with a better quality version.

After this correction I recommend this paper to publication in Plants.

Answer: Thank you.

Reviewer 2 Report

1. Restructure the abstract and add more information.
2.    Results are not comprehensively written and can be elaborated.
3.    Discussion can be improved from examples for the literature and more references to relate the results obtained.
4.     Also update and replace old references with recent references
5. Check figures and their ligands.
6. Elaborate the discussion section with a correlation of the studies with your work.
7. Restructure and carefully edit the conclusion section.

Author Response

Answers to reviewers (itemized answers in blue)

Reviewer 2

Restructure the abstract and add more information.

Answer: We added some more details in abstract.

Results are not comprehensively written and can be elaborated.

Answer: Some explanations were added in the results.

Discussion can be improved from examples for the literature and more references to relate the results obtained. Also update and replace old references with recent references. Elaborate the discussion section with a correlation of the studies with your work.

Answer: Five recent references (from 2021) were added to the list. They are correlating our work and our results with other studies on woody species in other regions. Additionally, several errors have been corrected.

Check figures and their ligands.

Answer: All figures and legends (as well as tables) have been checked and errors corrected. Additionally, the Figure 3 was changed with a better resolution version.

Restructure and carefully edit the conclusion section.

Answer: The conclusion section has been improved.
